# RPA activates the XPF-ERCC1 endonuclease to initiate processing of DNA interstrand crosslinks

Ummi B Abdullah[1], Joanna F McGouran[2,†], Sanja Brolih[1], Denis Ptchelkine[1,3], Afaf H El-Sagheer[2], Tom Brown[2,4] & Peter J McHugh[1,*] 

## Abstract

During replication-coupled DNA interstrand crosslink (ICL) repair, the XPF-ERCC1 endonuclease is required for the incisions that release, or "unhook", ICLs, but the mechanism of ICL unhooking remains largely unknown. Incisions are triggered when the nascent leading strand of a replication fork strikes the ICL. Here, we report that while purified XPF-ERCC1 incises simple ICL-containing model replication fork structures, the presence of a nascent leading strand, modelling the effects of replication arrest, inhibits this activity. Strikingly, the addition of the single-stranded DNA (ssDNA)-binding replication protein A (RPA) selectively restores XPF-ERCC1 endonuclease activity on this structure. The 5′–3′ exonuclease SNM1A can load from the XPF-ERCC1-RPA-induced incisions and digest past the crosslink to quantitatively complete the unhooking reaction. We postulate that these collaborative activities of XPF-ERCC1, RPA and SNM1A might explain how ICL unhooking is achieved *in vivo*.

**Keywords** Fanconi anaemia; interstrand crosslinks; RPA; SNM1A; XPF-ERCC1

**Subject Categories** DNA Replication, Repair & Recombination

**The EMBO Journal (2017) 36: 2047–2060**

See also: **D Klein Douwel et al** (July 2017) and **OD Schärer** (July 2017)

## Introduction

DNA interstrand crosslinks (ICLs) are cytotoxic DNA lesions that covalently tether the DNA double-helix inhibiting fundamental cellular processes that require DNA strand separation, including transcription and replication (Lawley & Phillips, 1996). Endogenously produced ICLs are believed to represent a major threat to genome integrity, illustrated by a rare inherited syndrome Fanconi anaemia (FA), associated with defective ICL repair (Duxin & Walter, 2015). FA patients suffer from bone marrow failure, predisposition

to solid tumours and numerous developmental defects (Brosh *et al*, 2017). ICL toxicity is also exploited in cancer chemotherapy, where the antiproliferative effects of a number of clinically important drugs (platinum agents, nitrogen mustards and mitomycin C) result from ICL induction (McHugh *et al*, 2001).

The major current model for vertebrate ICL repair was derived from studies of plasmids containing a site-specific ICL in cell-free *Xenopus laevis* egg extracts that support replication-coupled repair of ICLs (Raschle *et al*, 2008). The model proposes that the initial incisions of ICLs, a process termed "ICL unhooking", require the convergence of two replication forks upon the ICL. Both nascent leading strands initially stall ~20–40 nt from the ICL ("−20" position) due to steric hindrance imposed by the CMG helicase complexes (Fu *et al*, 2011). BRCA1-dependent eviction of the CMG helicase at the replication forks enables the subsequent extension of one nascent leading strand to immediately adjacent to the crosslinked nucleotides ("approach" to "−1" position; Long *et al*, 2014). Subsequently, ubiquitylated FANCD2-FANCDI facilitates the cleavage flanking the ICL by a nuclease(s) resulting in "unhooking" of the ICL. Uncoupling of the sister chromatids enables lesion bypass by polymerase Rev1 and extension past the lesion in Pol ζ-dependent manner (Raschle *et al*, 2008). The intact sister chromatid generated serves as a template for Rad51-dependent repair of residual DSBs via homologous recombination (Long *et al*, 2011).

At least six nucleases (XPF-ERCC1, MUS81-EME1, SNM1A, SNM1B, FAN1 and SLX1) have been implicated in the ICL unhooking step, because cells deficient in these proteins are hypersensitive to ICL-inducing agents and exhibit pathological responses to replication fork arrest at ICLs (De Silva *et al*, 2000; Niedernhofer *et al*, 2004; Bhagwat *et al*, 2009). Several lines of investigation suggest that XPF-ERCC1, a structure-selective heterodimeric endonuclease (Park *et al*, 1995; Sijbers *et al*, 1996), is likely to be an essential (although not necessarily sufficient) component of the ICL unhooking apparatus. XPF- or ERCC1-deficient mammalian cells are uniquely hypersensitive to ICL-inducing agents, compared to cells defective in other factors involved in nucleotide excision repair (NER), and suffer replication fork collapse upon ICL induction (De

1  Department of Oncology, Weatherall Institute of Molecular Medicine, University of Oxford, Oxford, UK
2  Department of Chemistry, University of Oxford, Oxford, UK
3  Research Complex at Harwell, Rutherford Appleton Laboratory, Oxford, UK
4  Department of Oncology, University of Oxford, Oxford, UK
   *Corresponding author. Tel: +44 1865 222441; E-mail: peter.mchugh@imm.ox.ac.uk
   †Present address: Department of Chemistry, Trinity College Dublin, Dublin, Ireland

Silva *et al*, 2000; Niedernhofer *et al*, 2004; Wang *et al*, 2011). Furthermore, the *Xenopus* cell-free replication-coupled repair system demonstrated that ICL unhooking is abolished when XPF is immunodepleted, but not MUS81 or FAN1 (Klein Douwel *et al*, 2014). Additionally, *in vitro* reconstitution assays have shown that XPF-ERCC1 alone is able to perform dual incisions flanking an ICL located near the ssDNA/dsDNA junction of a splayed arm "fork-like" (simple fork) DNA substrate, or in conjunction with the nuclease scaffold protein SLX4 (Kuraoka *et al*, 2000; Hodskinson *et al*, 2014; Klein Douwel *et al*, 2014), where SLX4 plays a key role in recruiting and positioning XPF-ERCC1 for incision (Klein Douwel *et al*, 2017). Finally, in the context of double-stranded DNA substrates, XPF-ERCC1 has been reported to act in an 3′-to-5′ exonuclease-like fashion, digesting past a site-specific ICL (Mu *et al*, 2000).

Despite recent advances in our understanding of the molecular mechanisms of ICL repair, it remains to be determined how XPF-ERCC1 processes the structures that arise during replication-coupled ICL repair (Deans & West, 2011; Clauson *et al*, 2013). Here, we examined the activity of purified human XPF-ERCC1 on DNA substrates that model native and ICL-damaged replication forks. Consistent with previous reports, XPF-ERCC1 incises simple fork structures, containing ICLs at their junction, within the duplex DNA region several nucleotides 5′ to the fork junction. However, the presence of a nascent leading strand, modelling the structure that triggers ICL incision, abrogates XPF-ERCC1 activity. Strikingly, the addition of the replicative single-stranded DNA (ssDNA)-binding replication protein A (RPA) selectively permits XPF-ERCC1 to overcome the inhibition by this structure. We then determined that the 5′-3′ ICL repair exonuclease SNM1A can load from XPF-ERCC1-RPA-induced incisions and digest past the ICL to complete ICL unhooking. We postulate that the collaborative efforts of XPF-ERCC1, RPA and SNM1A might explain how ICL unhooking is achieved *in vivo*.

# Results

## Nascent leading strands on replication fork structures inhibit XPF-ERCC1 activity

Current models for replication-coupled ICL repair indicate that the arrival of a nascent leading strand at the replication fork junction triggers ICL unhooking (Raschle *et al*, 2008). Therefore, we investigated the effect of model nascent leading and/or lagging strands on XPF-ERCC1 fork-processing activity through biochemical reconstitution analysis with purified XPF-ERCC1 (purification and enzyme activity validation shown in Fig EV1A–C). Consistent with previous reports (Kuraoka *et al*, 2000), XPF-ERCC1 (40 nM) incises approximately ~80% of a 3′-end-labelled simple fork substrate in 60 min, cutting at two major positions: six nucleotides (nt; ~50% of substrate) and two nucleotides (nt; ~27% of substrate) from the fork junction (Fig 1A–C, location of incisions confirmed through 5′-end-labelling in Fig EV2), giving 29-mer and 25-mer products, respectively. However, at the same XPF-ERCC1 concentration, only ~5% of a fork substrate bearing a model nascent leading strand (5′-flap structure, here denoted as "+leading-strand" structure) is incised, in line with previous reports that Rad1-Rad10 (the

budding yeast homologues of XPF-ERCC1) incisions are inhibited on a 5′-flap structure (Rodriguez *et al*, 1996; Fig 1A–C, confirmed with 5′-labelled substrates in Fig EV2). Moreover, on a fork structure containing a model nascent lagging strand (3′-flap structure, denoted as "+lagging-strand" structure) or model fork with both nascent leading and lagging strands, we did not detect XPF-ERCC1 incisions, consistent with the reported inhibition of XPF-ERCC1 on 3′-flap structures versus simple fork structures (Rodriguez *et al*, 1996; de Laat *et al*, 1998a; Figs 1A and EV2). We also confirmed that as expected, only the strand with a 3′-flap structure (the labelled strand in Figs 1A and EV2) is incised by XPF-ERCC1, for all structures tested (Fig EV3A). To emulate the dynamic nature of replication fork progression *in vitro*, we generated a fork substrate annealed to increasing lengths of model nascent leading strand, denoted: −9, −3, −2, −1 and 0 nt from the fork junction (Fig 1D). Each substrate represents a snapshot, modelling the gradual extension of the leading strand to the fork junction *in vivo*. XPF-ERCC1 activity gradually decreases as the leading strand extends closer to the junction (Fig 1D–F). Furthermore, XPF-ERCC1 incision is also gradually shifted and focused to a single major position 6 nt from the fork junction, with loss of the incision 2 nt from the junction (Fig 1D–F), likely due to steric effects. These data further confirm that the approach of a leading strand has a major inhibitory effect on XPF-ERCC1 activity, even though ICL incision is triggered by the arrival of a nascent leading strand at the replication fork junction (Raschle *et al*, 2008).

Since our data indicate that XPF-ERCC1 alone does not have the capability to incise the DNA upon the arrival of a leading strand at the fork junction, we hypothesised that additional factors may be required to facilitate XPF-ERCC1 incision(s).

## Inhibition of XPF-ERCC1 by a model nascent leading strand is overcome by RPA

RPA is a well-established XPF-interacting factor, required to position and activate XPF-ERCC1 for incisions made 5′ to the lesion during nucleotide excision repair (Matsunaga *et al*, 1996; Bessho *et al*, 1997; de Laat *et al*, 1998b). To explore the effect of RPA on XPF-ERCC1 activity, a simple fork substrate was pre-incubated with increasing concentrations of recombinant human RPA on ice for 10 min prior to reaction with 40 nM XPF-ERCC1 for 60 min at 30°C. Increasing levels of substrate incision were observed with increasing concentration of RPA, with near-quantitative incision (> 95%) observed at 80 nM RPA, a twofold excess over XPF-ERCC1 (Fig 2A and B). Furthermore, the presence of RPA shifted and focused XPF-ERCC1 incisions to a single site further into the duplex region (5′), away from the fork junction, with 95% of incisions 6 nt occurring from the junction (giving a 29-mer product) and loss of incision 2 nt from the junction (25-mer product; Fig 2A–C). By contrast, when the same assay was conducted using *Escherichia coli* single-strand DNA-binding protein (SSB), XPF-ERCC1 was inhibited, implying that the stimulatory effect on XPF-ERCC1 is RPA-specific (Fig 2A). Consistent with RPA requiring between 20 and 30 nucleotides to bind regions of ssDNA in its high-affinity mode (Wold, 1997), reducing the length of the forks arms to 13 nt eliminated the stimulatory effect of RPA on XPF-ERCC1 (Fig EV3B), with the caveat that XPF-ERCC1 was substantially less active on the shorter-armed substrate on its own.

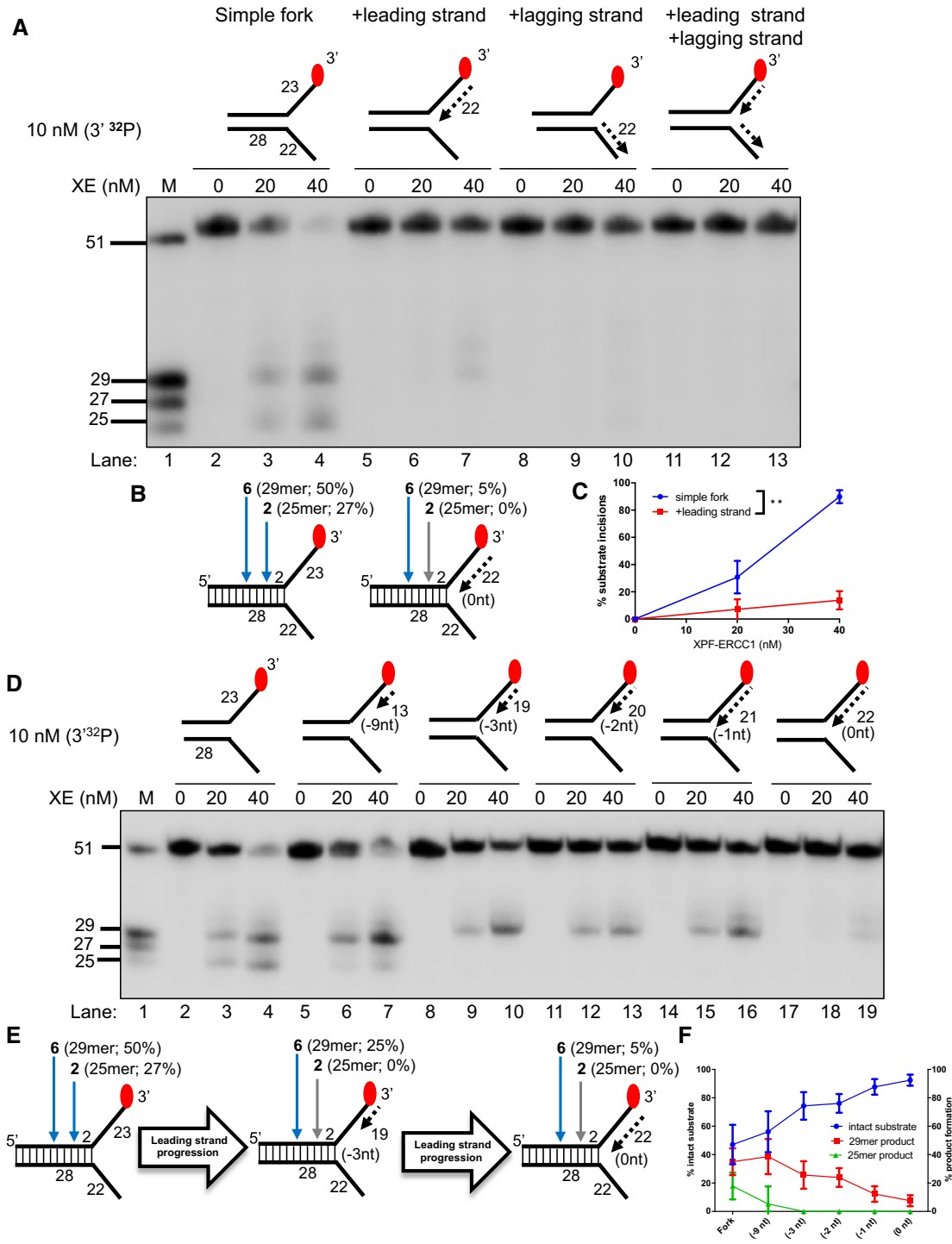

**Figure 1. Nascent leading strands on replication fork structures inhibit XPF-ERCC1 (XE) activity.**

A   Nuclease activity of XE on "fork-like" DNA substrates (simple fork; +leading strand; +lagging strand; +leading and +lagging strands). The red circles denote 3′[32P]-radiolabelled nucleotides. "M" denotes molecular weight marker.

B   A schematic representation of XE incisions on a "simple fork" and "+leading-strand" substrates. The positions of incision with respect to the fork junctions are indicated in bold (2 or 6 nt from the fork junction), and size of 3′[32P]-labelled incision products and percentage of incisions are indicated in parentheses.

C   Quantification of total substrate incisions expressed as a percentage of initial substrate as in (B). Unpaired two-tailed *t*-test; **P < 0.01. Error bars represent SEM, *n* = 3.

D   Nuclease activity of XE on fork substrates with increasing length of a model nascent leading strand (simple fork; −9; −3; −2; −1; 0 nt from the fork junction).

E   A schematic representation of XPF-ERCC1 incisions as in (D).

F   Quantification of intact substrate and incision products expressed as a percentage of initial substrate at 40 nM XE as in (D). Error bars represent SEM, *n* = 3.

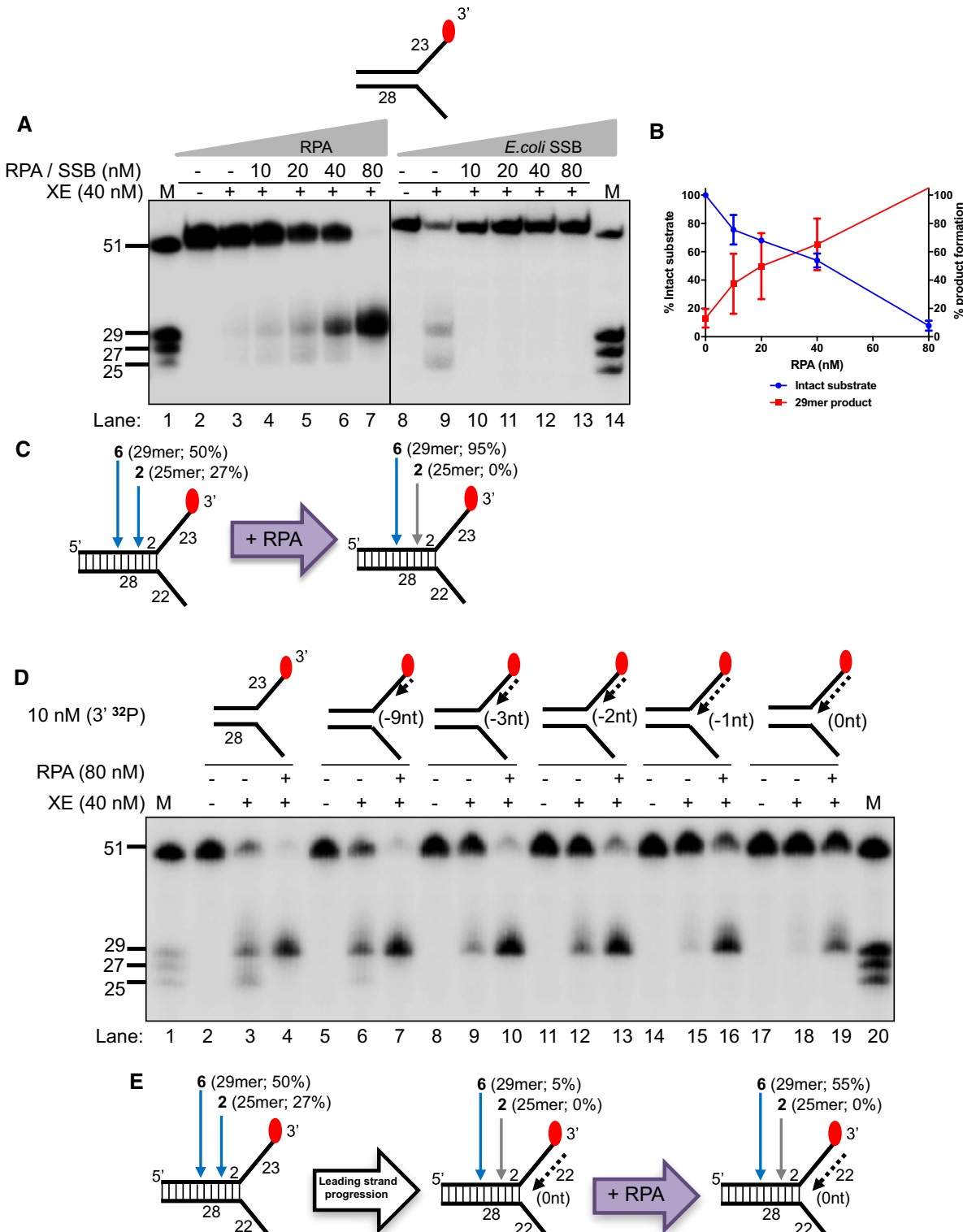

**Figure 2. Inhibition of XE activity by a model nascent leading strand is overcome by the addition of RPA.**

A  Nuclease activity of XE on a "simple fork" with increasing concentration of RPA or SSB (*Escherichia coli*).

B  Quantification of intact substrate and incision products as a percentage of initial substrate as in (A). Error bars represent SEM, *n* = 3.

C  A schematic representation of 40 nM XE activity on a "simple fork" in the absence (left) or presence (right) of 80 nM RPA.

D  Nuclease activity of XE on fork substrates with increasing length of a model nascent leading strand (simple fork; −9; −3; −2; −1; 0 nt from the fork junction) in the presence or absence of 80 nM RPA.

E  A schematic representation of XE activity in the presence and absence of RPA as in (D).

We also asked whether RPA modulates XPF-ERCC1 activity on the fork substrates that model the stepwise convergence of a model leading strand (Fig 2D). In the presence of RPA, XPF-ERCC1 activity increased on all substrates, confirming that RPA is not only able to stimulate XPF-ERCC1 activity on a simple fork substrate, but it is also required to permit XPF-ERCC1 activity in the presence of a model leading strand at various positions relative to the fork junction. Again, as the leading fork approaches the incision further into the duplex (5′), 6 nt from the fork junction (giving a 29-mer product) predominates.

We next determined whether the stimulation of XPF-ERCC1 by RPA is specific for fork substrates containing a model leading strand. Again, RPA stimulated XPF-ERCC1 on a simple fork substrate and on a fork substrate containing a model leading strand. However, RPA failed to restore XPF-ERCC1 activity on fork substrates containing either a model nascent lagging strand or both model nascent leading and lagging strands (Fig EV4A). The simultaneous presence of both leading and lagging strands on a fork substrate will prevent the association of RPA with the substrate, due to the absence of any ssDNA character, and therefore, the absence of XPF-ERCC1 activation on these substrates might be expected. However, the lack of stimulation on the substrate containing a model nascent lagging strand suggests that the stimulation of XPF-ERCC1 by RPA is selective for fork structures containing an unoccupied ssDNA 5′-flap region.

### RPA must bind a 5′-ssDNA flap to stimulate XPF-ERCC1 activity

We employed several approaches to confirm the importance of the presence of a 5′-ssDNA flap in the stimulation of XPF-ERCC1 by RPA at fork structures. First, we measured the affinity of RPA for the 5′-ssDNA flap region in the presence and absence of model leading strand and lagging strands using fluorescence anisotropy (Figs 3A and EV4B). In keeping with reported data on numerous other ssDNA-containing substrates (Kim *et al*, 1992, 1994), we found the $K_D$ of RPA for the simple fork, but also the fork containing a model leading strand to be close to 6 nM (Fig 3A). Interestingly, the affinity for the model lagging-strand substrate was also very similar to the simple model fork ($K_D$ = 6.2 nM, Fig EV4B), indicating that the orientation/polarity of RPA, rather than simply the affinity of RPA on the fork substrate, might be critical for guiding incisions to the leading-strand template. This is in keeping with previous reports of RPA having an orientation-specific effect on XPF-ERCC1 activity during the cleavage of hairpin substrates (de Laat *et al*, 1998b). We next ruled out the possibility that the selective stimulation of XPF-ERCC1 by RPA for structures containing a model leading strand could be the result of displacement of the leading strand at the RPA concentrations employed in our assays, producing a simple fork structure which acts as the reaction substrate (Fig EV4C). By labelling the nascent leading-strand molecule in our substrates, instead of the model fork template strand as in previous experiments, and incubating with 80 nM RPA, the RPA concentration employed in our previous experiments (Fig 2), followed by analysis on non-denaturing gels, we determined that the nascent leading strand-containing structure remains completely stable during incubation with 80 nM RPA for the reaction duration, ruling out this explanation for the mode of stimulation (Fig EV4C, lanes 4 and 9). To further establish the

importance of the 5′-ssDNA region, we generated substrates identical to those employed in Figs 1 and 2, except that the 5′-ssDNA flap region was replaced by a 5′-ssRNA flap (Fig 3B). The affinity of RPA for RNA is in the order of $10^{-3}$ to $10^{-4}$ fold less than for DNA (Kim *et al*, 1992). Consistent with the requirement for a 5′-ssDNA flap for the stimulation of XPF-ERCC1 by RPA, the RNA-containing substrate was not able to stimulate XPF-ERCC1 incision. We also examined a truncated form of RPA, RPA70ΔC442, that retains the high-affinity central ssDNA-binding domain of RPA70, DNA-binding domains (DBD) A and B but lacks the residues 442–661 located in the C-terminus of the 70-kDa subunit and 32- and 14-kDa RPA subunits (Lao *et al*, 2000). This form of RPA was unable to stimulate the XPF-ERCC1 activity, on either the simple fork or leading-strand substrate, confirming that full RPA trimer is required for the activation of XPF-ERCC1, not simply the association of RPA70 and DNA (Fig 3C). Finally, we determined whether the association of RPA with simple forks or leading-strand forks produced any conformational changes in the RPA heterotrimer, which could help account for its stimulatory role in fork incision upon DNA binding. Using limited tryptic digest, coupled to PAGE gel fragment analysis, we found that the association of RPA with either structure produced a marked, indistinguishable change in trypsin sensitivity, consistent with similar conformational changes being induced in RPA by both structures (Fig 3D). We conclude that a 5′-ssDNA flap is essential for the stimulation of XPF-ERCC1, that the full heterotrimeric form of RPA is also required, and that RPA undergoes an similar conformational change upon association with the 5′-ssDNA flap of simple fork structures, or those containing a model leading strand.

### The presence of an ICL at the fork junction inhibits XPF-ERCC1 incisions close to the fork junction

We next generated substrates that model replication fork collision with an ICL. The substrates contain a single site-specific triazole ICL placed at the fork junction (Figs 4A and EV5, Appendix Fig S1; Kocalka *et al*, 2008). We analysed the activity of XPF-ERCC1 on this substrate in the presence and absence of RPA and nascent model leading strand, initially employing 3′-end-labelled substrates. However, the triazole ICL is not heat labile making it difficult to precisely map sites of incision on these 3′-end-labelled substrates due to the limited mobility shift on PAGE gels (Fig 4B). We therefore analysed XPF-ERCC1 activity on 5′-radiolabelled ICL-containing fork substrates, which revealed a predominant 22-mer product, consistent with the major cleavage site being located 6 nt from the junction, and a second minor incision 9 nt from the junction (Fig 4A and C, lane 3). Notably, one of the major incisions observed in the control substrate that lacks an ICL, occurring 2 nt from the junction, is eliminated in the presence of the ICL, suggesting that the presence of this ICL at the junction inhibits nearby incision by XPF-ERCC1 (Fig 4C, lane 5). We next examined the consequences of a model leading strand on incision of this ICL-containing substrate. As for the native substrates, the presence of a model leading strand on the ICL-containing fork substrates inhibited XPF-ERCC1 activity (Fig 4D, lane 5). Once again, the addition of RPA allowed XPF-ERCC1 to overcome the inhibition conferred by the model leading strand (Fig 4D, lane 6).

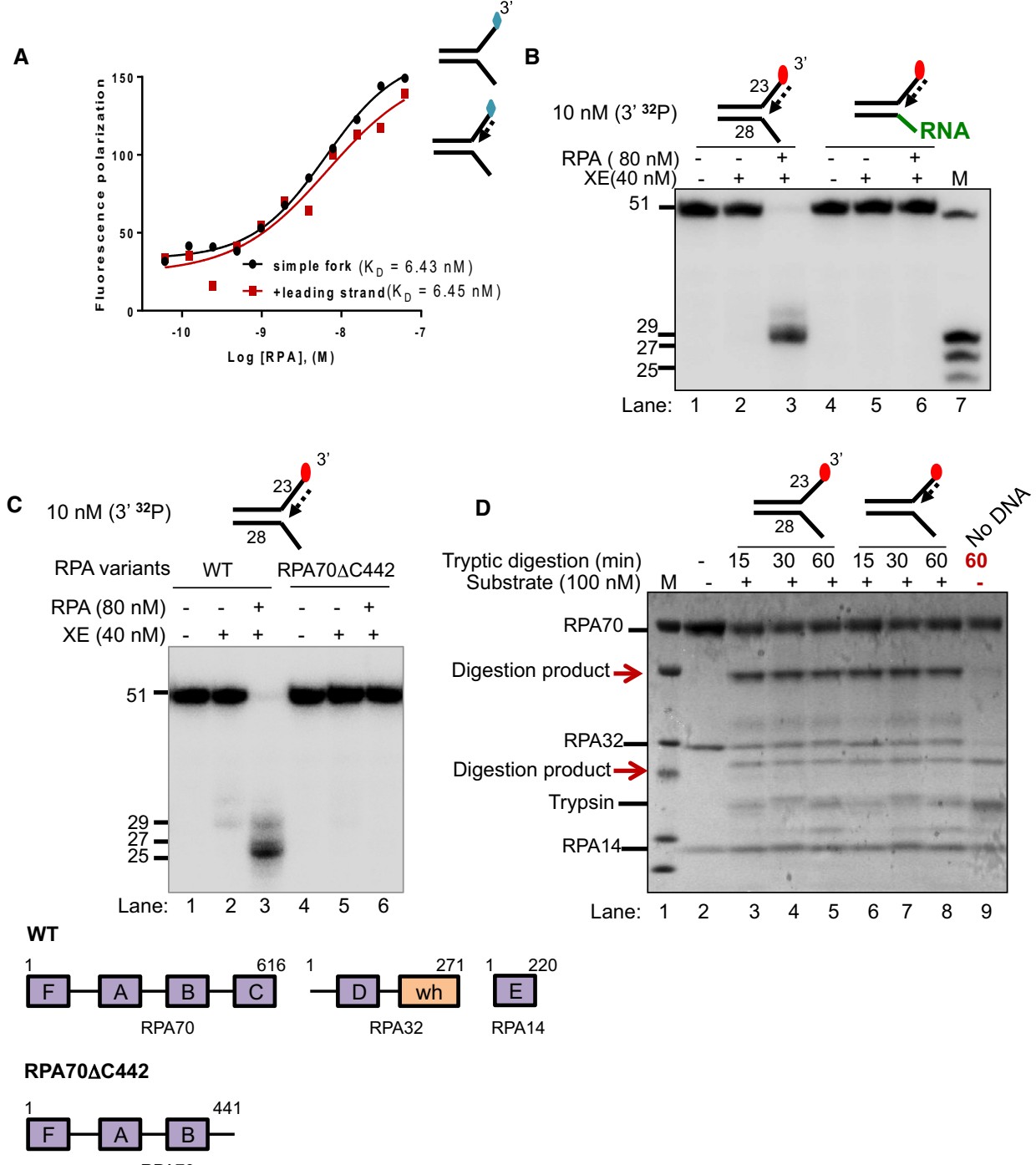

**Figure 3. RPA stimulates XPF-ERCC1 activity by binding to the 5′ arms of a DNA fork substrate.**

A   Fluorescence anisotropy assay to determine the binding constants of RPA for either "simple fork" or "+leading-strand" substrates. The blue diamonds denote the fluorophore-labelled nucleotides.

B   Nuclease activity of XE on "+leading strand" or DNA:RNA hybrid (5′ ssRNA on the bottom strand) substrates in the presence or absence of 80 nM RPA. RPA cannot stimulate XE to overcome the inhibition of a model nascent leading strand when the 5′-ssDNA overhang is replaced with 5′ ssRNA. Green line denotes RNA.

C   (Top panel) Nuclease activity of XE on "+leading-strand" substrate in the presence or absence of either the WT RPA or the truncated RPA (RPA70C442). (Bottom panel) A schematic representation of the structural domains of WT RPA and RPA70C442. Purple boxes represent the DNA-binding domains (DBD) designated as A–F. The orange box represents the winged helix domain. RPA70, RPA32 and RPA14 denote the three subunits of RPA.

D   Limited proteolysis assay to determine structural changes in RPA in the presence or absence of the indicated substrates. 800 nM RPA was incubated with 100 nM unlabelled DNA substrates (simple fork; +leading strand; or no DNA) prior to digestion with 500 nM trypsin in a time course. Reaction samples were separated in Bis-Tris SDS–PAGE (4–12%) and stained with InstantBlue. Red arrows indicated tryptic digestion pattern of RPA.

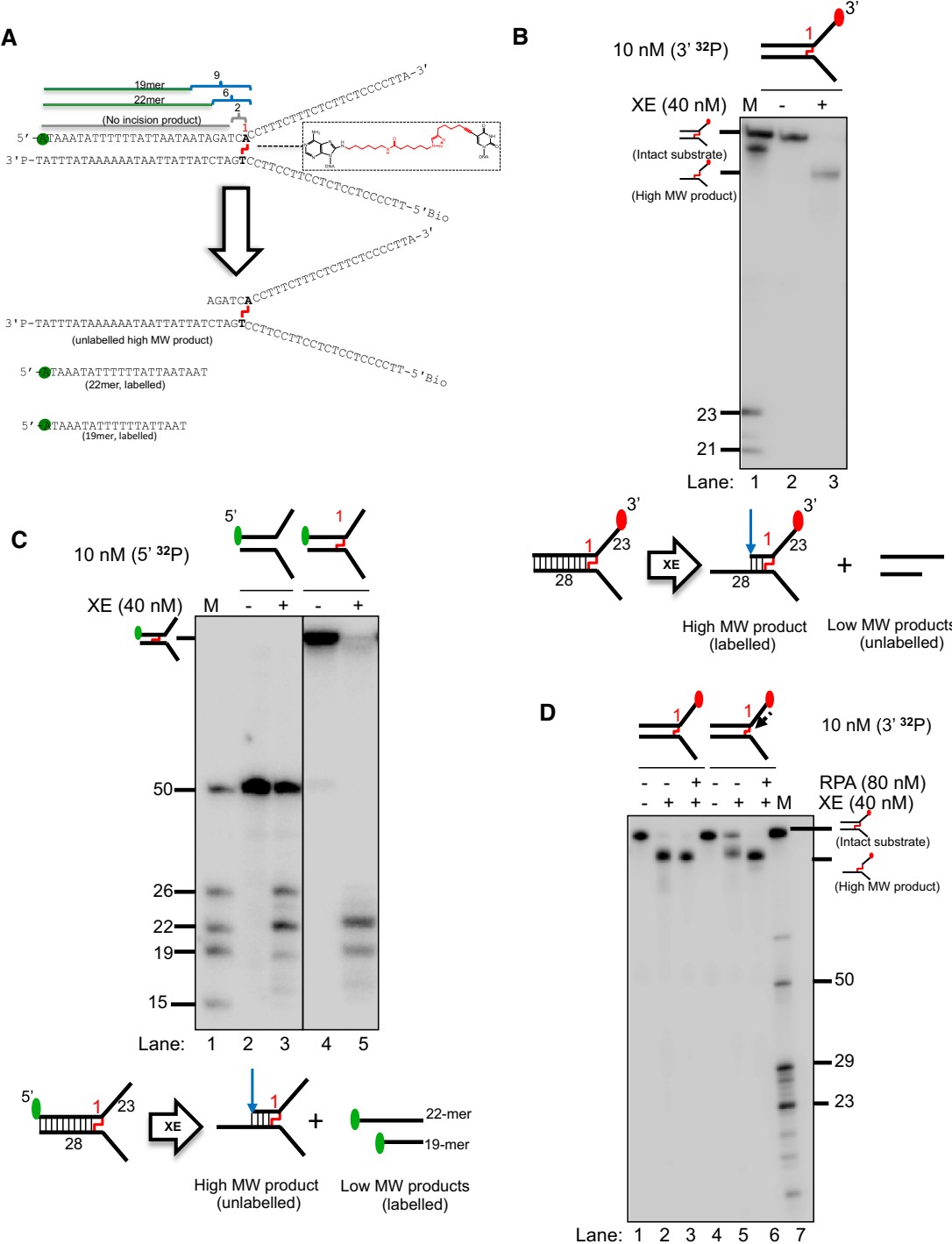

**Figure 4.  A model nascent leading strand also inhibits XE activity on a fork structure containing a single triazole interstrand crosslink (ICL), but the presence of RPA overcomes this inhibition.**

A    Sequence and schematic structure of a "simple fork" containing a single triazole ICL at the fork junction and its predicted XE nuclease incision products when radiolabelled on the 5′-end, based on the data obtained on non-crosslinked fork structure in Figs 1–4. Green circles denote 5′[$^{32}$P]-radiolabelled nucleotides.

B    (Top panel) Nuclease activity of XE on 3′[$^{32}$P]-labelled crosslinked simple fork substrate. (Bottom panel) A schematic representation of the nuclease reaction and the incision products.

C    (Top panel) Nuclease activity of XE on 5′[$^{32}$P]-labelled model native (lanes 2 and 3) and crosslinked (lanes 4 and 5) DNA substrates. The XE incision closest to the fork junction (2 nt from the junction, 26-mer product) is inhibited in the presence of a crosslink at the fork junction (lane 5). (Bottom panel) Schematic representation of the nuclease reaction and its incision products.

D    Nuclease activity of XE on 3′[$^{32}$P]-radiolabelled crosslinked substrate (simple fork; +leading strand) in the presence or absence of 80 nM RPA. XPF-ERCC1 incisions reduced by a leading strand are overcome by the presence of RPA (compare lane 5 to lane 6).

### SNM1A can load onto and digest past the ICL from an XPF-ERCC1-RPA-induced incision

Previous genetic and cellular studies revealed that XPF-ERCC1 collaborates with another repair factor, the 5′-to-3′ polarity exonuclease SNM1A, during replication-coupled ICL repair (Wang *et al*, 2011). Biochemical analysis of purified human SNM1A demonstrated that it loads onto and digests ICL-containing DNA substrates from either blunt ends, or, and of importance here, a single nick 5′ (upstream) to the ICL (Sengerova *et al*, 2012; Allerston *et al*, 2015). Therefore, we tested the capacity of SNM1A to digest native and ICL-containing fork substrates following XPF-ERCC1 incision (note: all the substrates used in Fig 5 contain 5′-hydroxyl terminal groups preventing exonucleolytic digestion from the substrate termini by SNM1A, which requires a 5′-phosphate group to initiate digestion). On a simple fork substrate, the reaction with XPF-ERCC1 alone produced the expected major incision products 2 and 6 nt into the duplex region (Fig 5A, lane 3). When the substrate is further incubated with SNM1A for increasing time, stepwise digestion products were observed consistent with its previously described 5′–3′ exonuclease activity. Within 30 min of incubation with SNM1A, digestion products shorter than the 23-mer marker were observed, indicating that SNM1A is able to digest past the fork junction and that its activity is not affected when the DNA substrate transitions from dsDNA to ssDNA (Fig 5A, lanes 2–9). Once again, when a model leading strand is present on the same substrate, XPF-ERCC1 incision is inhibited, which in turn prevents SNM1A from loading onto and digesting the DNA substrate (Fig 5A, lanes 10–13). As expected, the inhibition of XPF-ERCC1 activity by a leading strand on a fork substrate is restored by RPA. However, SNM1A was not able to digest the incised product to the point of the fork junction (23 nt; Fig 5A, lanes 14–17). This is plausibly the result of the increasingly limited base pairing between the SNM1A substrate strand (leading-strand template) and its complementary strand leading to lability as digestion approaches the junction, causing the dissociation of this substrate strand and interruption of the processing by SNM1A.

We next determined whether on the ICL-containing fork substrate, XPF-ERCC1-induced incision 5′ to the ICL enables digestion of the substrate from the incision site by SNM1A therefore releasing or "unhooking" the ICL (Fig 5B). The ICL-containing simple fork and leading-strand substrates were reacted with XPF-ERCC1 in the presence of RPA (XPF-ERCC1-RPA), and incubated with SNM1A for increasing times. On the simple fork, XPF-ERCC1-RPA incision allowed SNM1A to load and digest past the ICL, terminating approximately 4 nt after into the junction (the predominant products are between 23 and 19 nt). This reaction is sufficient to "unhook" the ICL. In the presence of a model leading strand, again, highly efficient XPF-ERCC1-RPA incision occurs. SNM1A was able to load in this substrate and progressively digest in the 5′–3′ direction, and continue to a position several nucleotides beyond the point of the junction (the major product is 19 nt; Fig 5B). This is in contrast to the non-crosslinked substrate utilised in Fig 5A, where SNM1A was unable to digest past the junction site. It appears that the covalent linking of the substrate strand to its complementary strand prevents dissociation of the SNM1A substrate strand facilitating digestion past the ICL. This reaction unhooks the ICL, and under the conditions used, this reaction is near-quantitative after 15 min incubation with SNM1A. This experiment reconstitutes an unhooking reaction that utilises a combination of endo- and exonuclease activities previously demonstrated to act within the same replication-coupled ICL repair pathway *in vivo*.

## Discussion

A number of major breakthroughs in our understanding of replication-coupled ICL repair have recently emerged from a cell-free replication-coupled repair system in X*enopus laevis* egg extracts (Raschle *et al*, 2008; Klein Douwel *et al*, 2014; Zhang *et al*, 2015). ICL repair is triggered by the convergence of dual replication forks at an ICL, and moreover, this convergence is a near-absolute requirement for the initiation of ICL repair (Zhang *et al*, 2015). One of the nascent leading strands eventually progresses to a position immediately adjacent (−1 position, equivalent to the "0 nt" substrate used in our analysis) to the ICL, and this triggers the ICL incision and processing reactions required for unhooking. Depletion of XPF-ERCC1 effectively abolishes ICL unhooking, but not the depletion of two other nucleases previously proposed to initiate this step, namely Mus81 and FAN1. Therefore, while XPF-ERCC1 is absolutely required for one of the initiating incisions during ICL unhooking, it might not be sufficient to achieve the complete unhooking reaction, where additional factors could contribute, perhaps redundantly.

Here, using an *in vitro* approach, we attempted to reconstitute ICL unhooking reactions using purified XPF-ERCC1 and a variety of synthetic DNA structures that mimic the ICL processing intermediates identified in the studies cited above. We discovered that, as expected, a simple splayed-arm/fork structure is a substrate for XPF-ERCC1 incision, the heterodimer cutting at several sites in the duplex region, between 2 and 6 nt from the junction. Addition of a model nascent leading strand, lagging strand or both strands was strongly inhibitory to the endonucleolytic activity of XPF-ERCC1. Consequently, we determined whether any known XPF-ERCC1-interacting factors relevant to replication-coupled ICL repair could modulate this reaction. Consequently, we investigated a very well-established XPF-ERCC1-interacting partner relevant to replication-coupled repair, RPA (Matsunaga *et al*, 1996; Bessho *et al*, 1997; de Laat *et al*, 1998b). Strikingly, for simple fork structures RPA was able to dramatically stimulate XPF-ERCC1 cleavage. Moreover, the presence of RPA efficiently overcame the inhibitory effect of the nascent leading strand, and we found that RPA has an equivalent affinity for fork structures whether or not a model nascent leading strand is present. However, stimulation was highly selective for substrates bearing a nascent leading strand as no rescue was observed on substrates containing a model nascent lagging strand or both model leading and lagging strands. This is despite the fact that RPA is able to interact with substrates bearing a model nascent lagging strand, confirming the importance of orientation/polarity of RPA in relation to its stimulation of XPF-ERCC1 activity. The absence of XPF-ERCC1 stimulation by RPA on fork substrates containing a nascent lagging strand implies that the simultaneous occupation of the 3′-flap region of fork structures by RPA and the 5′-flap region by a nascent lagging strand prevents XPF-ERCC1 cleavage, plausibly because the fork junction—the major feature recognised by XPF-ERCC1—is occluded. We also confirmed that the opposing 5′-DNA flap was essential to stimulate the RPA on native

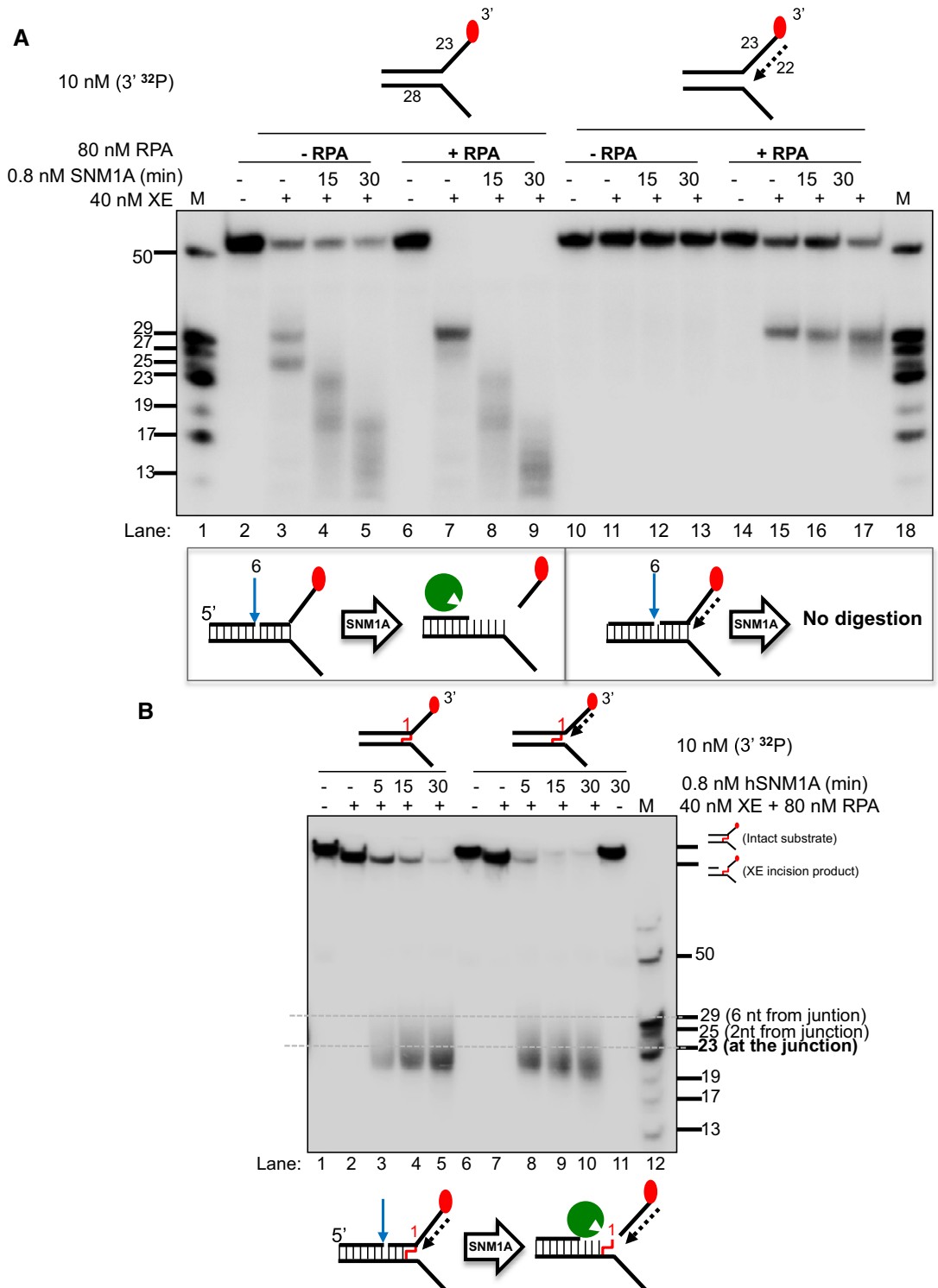

**Figure 5.  The 5′–3′ exonuclease SNM1A can load from an incision induced by XPF-ERCC1 to digest past a crosslink.**

A  (Top panel) Nuclease activity of XE on "simple fork" and +leading-strand substrates in the presence or absence of RPA and further incubated with 0.8 nM SNM1A in a time course. (Bottom panel) Schematic representation of the nuclease assay reaction products. The blue arrow denotes incision by XE and green Pacman represents digestion by SNM1A. RPA does not affect/alter SNM1A exonuclease activity as seen in the similar stepwise digestion products of SNM1A in the presence or absence of RPA (lanes 2–9). However, the presence of a model leading strand prevents SNM1A from loading onto XE-RPA-induced incisions to digest the DNA substrate (lanes 10–17).

B  (Top panel) Nuclease activity of XE-RPA on crosslinked DNA substrates (simple fork; +leading strand) and further incubation with SNM1A in a time course. SNM1A digestion inhibition by a model nascent leading strand (as in A) is overcome when an ICL is located at the fork junction.

forks and those containing a nascent leading strand by synthesising structures where the DNA of the 5′-flap region is swapped for RNA. Moreover, RPA associated with either simple forks or forks containing model nascent leading strands undergoes comparable conformational changes, providing an indication that structural alterations in RPA might play a role in activating XPF-ERCC1. The details of this mechanism of activation will require detailed structural and biophysical analysis, and are a key area to be addressed in future studies. Moreover, a minimal, XPF-interacting version of the SLX4 (mini-SLX4) has been reported to have a stimulatory effect on XPF-ERCC1 during the processing of simple fork structures with and without ICLs incorporated (Hodskinson *et al*, 2014), and it will be important to examine the effect of additionally adding SLX4 protein into the RPA-dependent reaction we have reconstituted.

Our observations allowed us to attempt biochemical reconstitution of ICL unhooking *in vitro*, employing a synthetic model of the key intermediates triggering ICL incision, a fork with a nascent leading strand arriving at the crosslinked junction. As for undamaged DNA substrates, the presence of a nascent leading strand was inhibitory to XPF-ERCC1, and once again RPA could overcome this inhibition permitting incisions within the duplex region of the substrate. However, no second incisions were observed at, or 3′ to, the crosslinked junction site, which would be required to unhook the ICL. Therefore, it appears likely that an additional activity (or activities) is required to complete the unhooking reaction. A strong candidate for such an activity is the SNM1A 5′–3′ ICL repair exonuclease (Wang *et al*, 2011). SNM1A has been demonstrated to participate in replication-coupled ICL repair, in the same pathway as XPF-ERCC1 based on the genetic and cellular phenotypes of cells lacking either or both factors (Wang *et al*, 2011). Moreover, SNM1A possesses a striking capacity to digest DNA-containing adducts on the substrate strand, including ICLs (Sengerova *et al*, 2012; Allerston *et al*, 2015). ICL digestion leaves a residual single-nucleotide adduct tethered to the opposing strand (Wang *et al*, 2011). Notably, this intermediate is readily detected in the *Xenopus* cell-free replication-coupled repair system as a substrate for the downstream, TLS-mediated bypass stage of ICL repair (Raschle *et al*, 2008). In our attempts to reconstitute ICL incision reactions that accurately reflect the repair of ICLs in a cellular context, the work presented here utilises an ICL at the point of collision with the arrested leading strand. This differs to a previous study employing a simple fork harbouring an ICL several nucleotides internal to the fork junction, where sequential (3′

followed by 5′) unhooking incisions that bracket and unhook the ICL were observed (Kuraoka *et al*, 2000). In addition to the differences in ICL location and the enzymology employed (here, RPA and SNM1A were also considered), the structure of the substrates used in the study of Kuraoka *et al* and those presented here are substantially different. The ICLs we have used produce relatively little distortion of the DNA (Kocalka *et al*, 2008), in contrast to the psoralen ICLs used by Kuraoka *et al* Together with the location of the ICL relative to the fork junction, this could also have a substantial impact on the position and efficiency of incisions.

Recently, the FAN1 nuclease has also been demonstrated to act in an exo-like fashion and is able to degrade past and release DNA substrates containing ICLs (Wang *et al*, 2014; Zhao *et al*, 2014; Pizzolato *et al*, 2015). Moreover, studies in fission yeast and more recently in mouse cells imply that FAN1 and SNM1A may play a redundant role in ICL repair, or contribute to different ICL repair sub-pathways (Fontebasso *et al*, 2013; Thongthip *et al*, 2016). We have performed pilot studies to define the action of FAN1 on the ICL-containing substrates utilised in this study. However, the combined endo- and exonuclease activities of FAN1 on these substrates following initial incision by XPF-ERCC1 yielded an extremely complex mixture of products, and it will require new approaches to fully understand the mode of action of FAN1 in combination with XPF-ERCC1, RPA and, potentially, SNM1A.

Although the experiments presented here were performed on substrates containing a single fork, they are informative regarding the likely events occurring during replication fork convergence, since due to the location of fork stalling, a region of approximately 20–40 nucleotides of annealed, double-strand DNA will persist between the converging forks (Raschle *et al*, 2008). In this regard, our work highlights a substantial difference between the current proposed models of ICL repair and a modified model consistent with our data (Fig 6; Raschle *et al*, 2008; Zhang & Walter, 2014). The fundamental predictions of our model are relevant to ICL repair at a single fork, or to the converging fork model. In previous models, the incisions triggered by arrival of the nascent leading strand at the ICL site (described as the −1 position in the *Xenopus* studies, position "0 nt" in our substrates) were postulated to occur on the lagging-strand template, rather than the leading-strand template, as observed here. It has not yet been possible to map incision locations during the repair in the *Xenopus* cell-free system, but we strongly predict that they will occur on the

---

**Figure 6. Model for the collaborative activity of XPF-ERCC1, RPA and SNM1A to unhook a crosslink.**

A    When a single replication fork encounters an ICL, the nascent leading strand initially stalls 20–40 nt from the ICL ("−20" position; step a-i). It gradually progresses to 1 nt from the ICL ("0" position; step a-ii), and its arrival at the ICL triggers an XPF-ERCC1-RPA-induced incision six nucleotides 5′ to the junction, in a duplex region (step a-iii). SNM1A loads from these incisions and digests past the ICL, unhooking the ICL from the DNA duplex, leaving a residual single nucleotide moiety (step a-iv), which has been demonstrated as the reaction product using mass spectrometry to characterise the reaction products of SNM1A activity in previous work (Wang *et al*, 2011). This enables translesion synthesis to occur and repair of the broken DNA strand via homologous recombination (step a-v).

B    In the event of dual replication fork convergence onto an ICL, both nascent leading strands initially stall ~20–40 nt from the ICL (step b-i). CMG complexes from both replication forks unload from both leading strands, as previously described (Long *et al*, 2014; Zhang *et al*, 2015) which enables one nascent leading strand to gradually progresses to 1 nt from the ICL ("0" position; step b-ii) as previously described (Raschle *et al*, 2008; Zhang *et al*, 2015). The structure that arises at this stage is inhibitory for XPF-ERCC1. However, in the presence of RPA, XPF-ERCC1 will be able to incise the structure (on the lagging-strand template associated with the fork which has progressed to 0 nt) within the duplex region, 6 nt from the ICL (step b-iv). This XPF-ERCC1-RPA-induced incision enables SNM1A to load onto and digest past the ICL (step b-v). The net result of XPF-ERCC1-RPA-SNM1A is ICL unhooking, which enable the translesion (TLS) synthesis step, where the strand extended by the TLS polymerase is the nascent leading strand which remained arrested at ~20–40 nt from the ICL on the second converged fork and did not strike the ICL (step b-vi). Homologous recombination-based repair of the broken chromatid completes repair and facilitates fork restart.

Data information: Black dotted arrows represent initial approach by nascent leading strands. Blue arrows represent incisions by XPF-ERCC1; green dotted arrows represent digestion by SNM1A; maroon dotted arrows represent nascent leading strand progression.

---

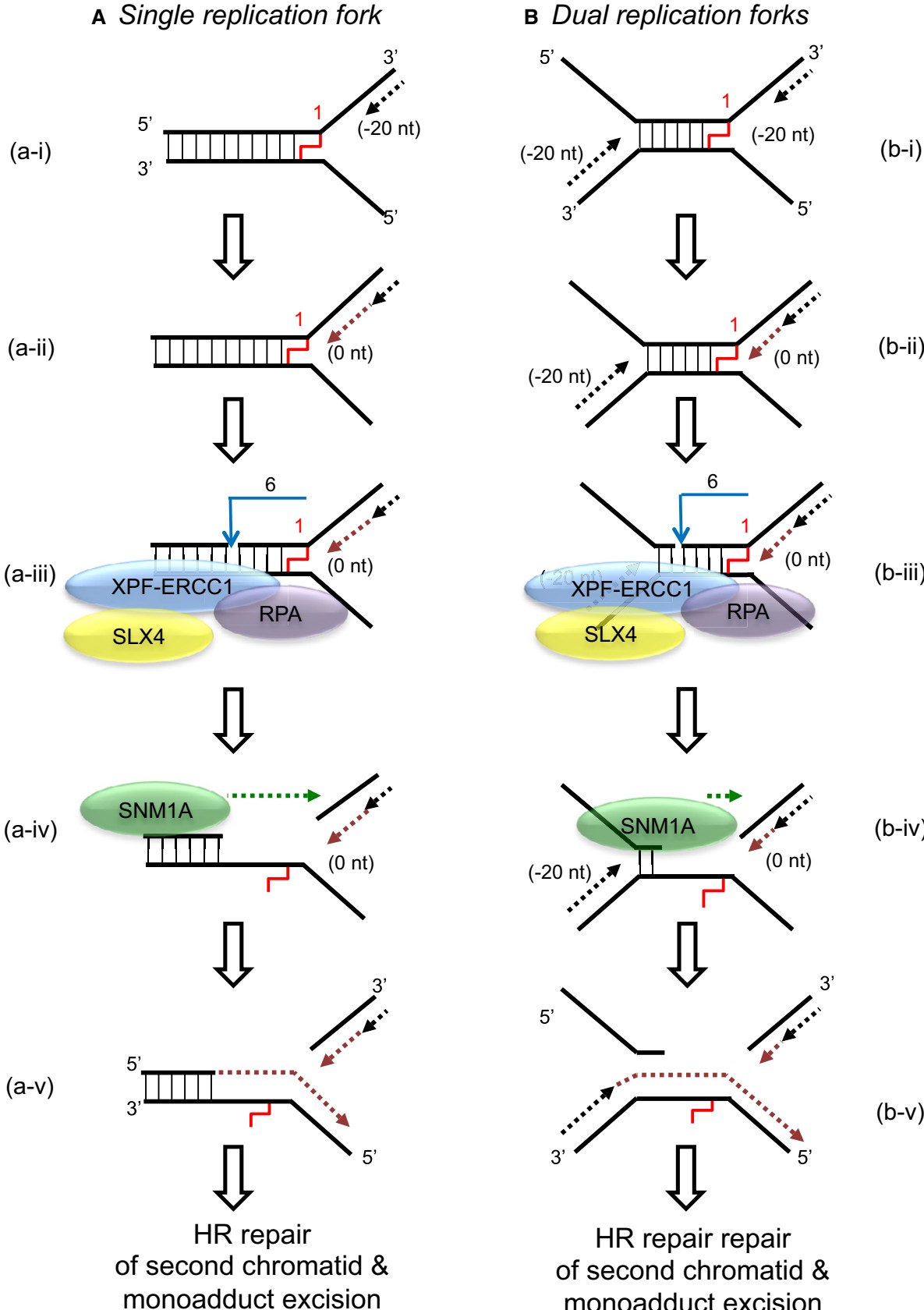

**Figure 6.**

leading-strand template. Following exonucleolytic digestion by SNM1A or FAN1, or additional factors to be identified, ICL unhooking can be completed providing a substrate where one or several nucleotides remain tethered to un-incised strand, and it is this unbroken strand that will be bypassed in the subsequent TLS (Pol ζ-dependent) stage of ICL repair (Budzowska *et al*, 2015). Based on our experimentally determined (on the putative lagging-strand template) location of the XPF-ERCC1 incisions, we predict that the strand that will be extended during TLS must be that nascent leading strand previously arrested at the −20 to −40 position from the ICL (Fig 6). Further studies in the *Xenopus* cell-free system will be required to confirm this.

Another ICL repair model has been proposed from results obtained in DNA combing experiments. The data support a model whereby replication forks "traverse" ICLs (possibly mediated by MCM helicase remodelling, or recruitment of an alternative helicase) leaving an ICL-containing X-structure (Huang *et al*, 2013). This structure would then be repaired post-replicatively, and this could plausibly be initiated by XPF-ERCC1 incision, depending upon the nature of the remaining post-traverse structure. Moreover, the FANCM translocase is important for fork traverse, and FANCM is proposed to aid recruitment of RPA to ICLs (Huang *et al*, 2010). This might provide a permissive environment for XPF-ERCC1 incision. In this regard, testing the activity of XPF-ERCC1 and associated factors on the structures predicted to remain after ICL traverse would allow their potential role in this process to be explored.

In summary, we have biochemically reconstituted a mechanism of ICL unhooking that is fully consistent with recent studies of ICL repair in *Xenopus* cell-free extracts and genetics and cellular studies in mammalian cell systems: the reaction represents plausible way in which this critical step of ICL repair is achieved. Moreover, our work reveals several key details of the unhooking process, defining the likely site(s) of incision, the nature of the unhooked intermediate and consequently defining the orientation of the downstream TLS step.

## Materials and Methods

### Purification of human recombinant XPF-ERCC1 from insect cells

Purification of XPF-ERCC1 protein complex achieved via modification of a published protocol (Enzlin & Scharer, 2002). Briefly, pFastBac1 vectors with the cDNA encoding the full-length XPF and the full-length ERCC1-His, a kind gift from O. D. Scharer (Stony Brook University, USA), were individually transformed into competent DH10Bac *E. coli* cells. Bacmid DNA was isolated and transfected into Sf-21 insect cells to amplify the baculovirus according to the manufacturer's instructions (BAC-To-BAC, Invitrogen). For protein production, 500 ml Hi-5 insect cell culture at the concentration of $2.5 \times 10^7$ cells/ml was co-infected with XPF and ERCC1 viruses. Cells were harvested 65 h post-infection and centrifuged at $400 \times g$ for 10 min at room temperature (RT). Cells were gently re-suspended in 200 ml 100 mM NaCl and centrifuged at $400 \times g$ for 10 min at RT and re-suspended in 100 ml lysis buffer (50 mM HEPES pH 8.0, 800 mM NaCl, 5 mM TCEP, 10% glycerol, 0.1% NP-40 and one dissolved EDTA-free protease inhibitor cocktail tablet (Roche)). Cell lysate was sonicated for six

cycles (1-min sonication, 2-min rest) with 9.5-mm probe using Soniprep 150 Plus (MSE). Cell lysate was clarified by centrifugation at 40,000 *g* for 1 h at 4°C. The supernatant were incubated with 2.5 ml Ni-NTA agarose beads (Qiagen) overnight on a rotating platform at 4°C.

The beads were collected by centrifugation at $400 \times g$ for 5 min at 4°C and re-suspended in 20 ml Ni-buffer (50 mM HEPES pH 8.0, 10% glycerol, 5 mM TCEP) containing 800 mM NaCl/4 mM imidazole and packed in a column. The column was washed twice via gravity flow with 50 ml Ni-buffer containing 500 mM NaCl/5 mM imidazole and 500 mM NaCl/20 mM imidazole, respectively. The XPF-ERCC1 complex was eluted twice with 12 ml Ni-buffer containing 300 mM NaCl/50 mM imidazole and once in 1 ml fractions with 12 ml Ni-buffer containing 300 mM NaCl/250 mM imidazole. The eluted fractions were loaded into dialysis cassettes with a 3.5 K molecular weight cut-off (Thermo Scientific) and dialysed overnight in 2 l dialysis buffer (50 mM Tris pH 8.0, 10% glycerol, 500 mM NaCl, 5 mM DTT). The XPF-ERCC1 heterodimer was further purified by gel filtration using a HiLoad 16/60 Superdex 200 column (Pharmacia) equilibrated with gel filtration buffer (25 mM HEPES pH 8.0, 150 mM NaCl, 10% glycerol and 5 mM TCEP). Fractions containing XPF-ERCC1 were pooled and concentrated. The protein concentration was measured directly at 280 nm, and the purified protein was flash-frozen in aliquots and stored at −80°C. Typically, 0.25–1 mg/ml of purified complex, with a concentration of 0.1–0.6 mg/ml, was obtained.

Full-length human RPA purified from *E. coli* was a kind gift from Fumiko Esashi (Dunn School of Pathology, University of Oxford). Truncated human RPA (RPA70C442) was a kind gifts from Marc Wold (University of Iowa, USA). Human SNM1A truncated for its first 675 residues (aa 676-1040) was purified using published protocol (Allerston *et al*, 2015). Full-length *E. coli* SSB was purchased from Sigma Aldrich.

### Synthesis, structure and sequence of DNA substrates

These are described in detail in the Appendix: Appendix Table S1, Appendix Figure S1 and Appendix Supplementary Methods.

### Generation of radiolabelled DNA substrates

For $3'[^{32}P]$-radiolabelling, 25 μl reaction containing 50 pmoles of DNA oligonucleotide, 1 unit of terminal deoxytransferase (TdT; NEB) and 3.3 pmoles $[\alpha-^{32}P]dATP$ was incubated for 1 h at 37°C. For $5'[^{32}P]$-radiolabelling, 25 μl reaction containing 10 pmoles of DNA oligonucleotide, 1 unit T4 polynucleotide kinase (T4 PNK; NEB) and 6.8 pmoles $[\gamma-^{32}P]dATP$ was incubated for 1 h at 37°C. The radiolabelled reaction mixture was loaded in a Bio-Gel P-6 spin column (Bio-Rad), centrifuged at $1,000 \times g$ for 4 min and diluted with nuclease-free water to the concentration of 500 nM. The radiolabelled oligonucleotide was annealed with unlabelled oligonucleotide(s) by boiling at 95°C for 5 min and gradually cooled to below 30°C in annealing buffer (10 mM Tris pH 7.5, 50 mM NaCl, 1 mM EDTA). The radiolabelled substrate is diluted to the working concentration of 100 nM. 1 μl of 100 nM DNA is used in each reaction. The quality of every batch of radiolabelled DNA substrates generated is analysed using 10% non-denaturing polyacrylamide (PAGE) gel. Gels were imaged and analysed.

## Nuclease assays

Unless otherwise stated, nuclease assays were carried out in a 10-μl reaction mixture containing 10 nM radiolabelled DNA substrate and 40 nM (62 ng) XPF-ERCC1 in nuclease buffer (25 mM HEPES pH 8.0, 40 mM NaCl, 10% glycerol, 0.5 mM β-mercaptoethanol, 0.1 mg/ml BSA) containing 0.4 mM $MnCl_2$. To analyse the nuclease activity of XPF-ERCC1, DNA substrates and XPF-ERCC1 proteins were incubated at 30°C for 60 min. To analyse the effect of RPA on the nuclease activity of XPF-ERCC1, DNA substrates were pre-incubated with 80 nM RPA (92.8 ng) on ice for 10 min followed by the addition of XPF-ERCC1. The reactions were incubated at 30°C for 60 min. To analyse the activity of hSNM1A post-incision by XPF-ERCC1, DNA substrates were reacted with XPF-ERCC1 (in the absence or presence of RPA) at 30°C for 60 min in nuclease buffer containing 10 mM $MgCl_2$, followed by the addition of 0.8 nM (1 μl of 8 nM) hSNM1A. The reactions were further incubated at 37°C for increasing time.

Nuclease assays were quenched by adding 3 μl stop solution (95% formamide/5% EDTA) and heating at 95°C for 5 min. Samples were loaded onto 10% or 20% denaturing (7 M urea) PAGE gel (19:1 acrylamide/Bis) gels containing 1× TBE and run for 2 h at 525 V. Gels were fixed in fixing solution (40% methanol/20% acetic acid/5% glycerol) for 60 min and dried in a gel dryer at 50°C for 4 h or 80°C for 2 h. Reaction products were visualised and quantified by Typhoon (GE Healthcare) phosphorimager. Unless otherwise stated, all nuclease assays are representative of experiments performed at least three times.

Line graphs showing percentage of substrate incisions represent decay of the substrate band quantified and expressed as a percentage of initial substrate. Line graphs showing per cent formation represent quantification of intact substrate and incision products as a percentage of initial substrate. Unpaired two-tailed *t*-test; *$P < 0.05$. Error bars represent SEM, $n = 3$.

## Fluorescence anisotropy

RPA was serially diluted twofold (125 μM to 61 pM) and then mixed 1:1 with 10 nM fluorescein-labelled DNA substrates in nuclease buffer containing 1 mM $MnCl_2$ in 30 μl reaction volume. Reactions were incubated at RT for 10 min and analysed using PHERAstar (BMG). Data were fitted using a sigmoidal curve in GraphPad Prism.

## Limited proteolysis

800 nM RPA was incubated with 100 nM unlabelled DNA substrates in nuclease buffer containing 5 mM $CaCl_2$ in 10 μl reaction volume at 30°C for 60 min and then digested with 500 nM sequencing-grade modified trypsin (Promega) for increasing time (15, 30, 60 min). Reaction samples were quenched with SDS–PAGE loading buffer followed by boiling for 3 min. The digestion products were separated and visualised by Bis-Tris SDS–PAGE gel (4–12%) and InstantBlue.

## Western blot analysis

Western blot analysis was performed using anti-XPF (Abcam, ab73720) and anti-ERCC1 (Santa-Cruz Biotechnology, FL-297) antibodies.

**Expanded View** for this article is available online.

## Acknowledgements

We thank our laboratory member Sook Y. Lee and Opher Gileadi (Structural Genomics Consortium, Oxford) for the recombinant SNM1A; Fumiko Esashi (Dunn School of Pathology, University of Oxford) and Marc Wold (Iowa State University, USA) for the RPA and RPA70ΔC442, respectively; Hannah Baddock (WIMM), Hazel Aitkenhead and Joseph A. Newman (Structural Genomics Consortium, University of Oxford) for assistance with fluorescence anisotropy. P.J.M. was supported by MRC (Medical Research Council, grant MR/L007665/1). A.H.E.S. and T.B. were supported by BBSRC (Biotechnology and Biological Sciences Research Council, grant BB/J001694/1) BBSRC: sLOLA grant BB/J001694/1 "Extending the boundaries of nucleic acid chemistry" J.F.M. was supported by King Abdullah University of Science and Technology (KAUST) for a grant awarded to B. Nordén. U.B.A. was supported by the Malaysia's King Scholarship (Biasiswa Yang Di-Pertuan Agong).

## Author contributions

UBA and PJM designed the experiments. UBA performed the experiments and analysed the data with help of PJM and SB. JFM, AHES and TB synthesised the crosslinked substrates. JFM, DP and UBA purified the recombinant XPF-ERCC1 protein. UBA, JFM and PJM drafted and revised the manuscript, and all authors reviewed the manuscript and provided comments.

## Conflict of interest

The authors declare that they have no conflict of interest.

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
