## [Review Process File · The EMBO Journal]

Manuscript EMBO-2017-96664

RPA activates the XPF-ERCC1 endonuclease to initiate processing of DNA interstrand crosslinks

Umami B. Abdullah, Joanna F. McGouran, Sanja Brolih, Denis Ptchelkine, Afaf H. El-Sagheer, Tom Brown, and Peter J. McHugh

Corresponding Author: Peter McHugh, University of Oxford

Review timeline:

Submission date:	03 February 2017
Editorial Decision:	06 March 2017
Revision received:	13 April 2017
Editorial Decision:	15 May 2017
Accepted:	19 May 2017

Editor: Hartmut Vodermaier

Transaction Report:

1st Editorial Decision

06 March 2017

Thank you for submitting your manuscript on RPA stimulation of XPF-ERCC1 in ICL processing to The EMBO Journal. It has now been assessed by three referees, and I am pleased to inform you that all of them find your conclusions potentially important and generally well supported by the data. Pending satisfactory revision of a number of specific concerns, we shall therefore be happy to consider the study further for publication.

As you will see, while most of the referee points refer to clarifications, discussions and presentational issues, there are also a few experimental suggestions that should strengthen the final manuscript, in particular in referee 3's point 4, and in the related points 3 and 4 of referee 1 and point 3 of referee 3. In any case, in light of our single-revision-round policy it will be important to diligently respond to all points raised at this stage.

REFeree REPORTS

Referee #1:

The enzymology of interstrand cross-link repair has not been well worked out. This biochemical study from an excellent team of scientists nicely demonstrates that XPF-ERCC1, one of the major nucleases known to be involved in interstrand cross-link repair is inhibited at a replication fork bearing a leading strand. Such a structure is expected to occur when the leading strand synthesis runs into an interstrand cross-link. This inhibition is shown to be overcome by the presence of the trimeric single-strand binding protein, RPA. Mechanistic studies suggest that this stimulation of

XPF-ERCC1 incision is due to the binding of RPA70 subunit to the single-strand DNA end in the replication fork. They nicely show that the C-terminus of RPA70 is required. Using an elegant click chemistry approach to produce site-specific crosslinks at the site of the replication fork the authors then show that 1) XPF-ERCC1 cuts 6 nucleotides from the junction and this nick can serve as entry point for the SNM1A exonuclease that can process through the cross-link. The authors provide a plausible model for interstrand cross-link repair at a single or a dual replication fork that is consistent with all known data. This is an important contribution to the field and makes direct predictions about the role of these proteins during repair in a living nucleus, which is clearly beyond the scope of this elegant work. The authors should consider the following points:

1. The small diagrams that are given with the incision experiments are extremely helpful. But in Figures 1, 2 and 3 showing the positions of the two nick sites at either 6 or 2 bp from the fork with brackets is very confusing. Perhaps they can adopt the convention shown in Figure 5 and the length of the arrow can be proportional to the amount of cutting observed?
2. Figure 1 panel F, plotting the remaining product with the amount of incision product is helpful. But perhaps two Y axis could be shown, one indicating product remaining and the other incision products. The word "Formation" is very misleading as the uncut product is not forming.
3. The authors show a direct interaction of XPF-ERCC1 with RPA in their model in Figure 6. Data supporting whether this direct interaction is necessary or sufficient versus the need for RPA binding to the DNA would provide impact. One experiment is to vary the length of this single strand DNA of the free end in the replication fork and determine the minimum length necessary for the stimulation. This type of experiment would suggest that binding of RPA to the single strand region allows interaction with XPF-ERCC1. I believe RPA requires 6 bases. Having the minimal flap size would then allow a fluorescence anisotropy experiment of XPF-ERCC1 (in the absence of Mg to inhibit incision) in the presence and absence of RPA. Binding of XPF-ERCC1 should show a large increase in anisotropy. A less elegant approach would be to do a pull down showing that XPF-ERCC1 can only load on a cross-link replication fork with a leading strand, if RPA is added. This could be done with biotin-labeled DNA and SA beads.
4. Another nice experiment that would provide more support for their model, would be to mutate key residues of the XPF-ERCC1 or RPA that are suggested to be at the interface of their interaction site.

Referee #2:

In this paper, the McHugh and coworkers investigate the influence of the single-strand binding protein RPA on the ability of XPF-ERCC1 to unhook replication fork-blocking interstrand crosslinks. Taking a biochemical approach, using radiolabelled model replication fork structures, they find that the addition of a nascent strand (either leading or lagging) to a "simple fork" (or "splayed Y") structure inhibits cleavage of the fork. They show that increasing nascent strand length increases the inhibitory effect on XPF-ERCC1. Intriguingly RPA, which is known to interact with XPF-ERCC1 and to facilitate the DNA repair processes that involve XPF-ERCC1, overcomes the inhibitory effect of nascent leading DNA strands on fork cleavage by XPF-ERCC1. RPA does not overcome the inhibitory effect of a nascent lagging strand, however. RPA appears to bias the position of incisions by XPF away from the fork junction. Evidence is presented that an intact RPA trimer is required to promote XPF-ERCC1 activity towards forks with nascent strands. The authors go on to show that placing an ICL at the fork junction biases the position of incisions by XPF away from the fork junction. This incision creates a site for loading of the SNM1A nuclease which can exonucleolytically degrade DNA past the ICL, thereby unhooking to ICL.

This is an interesting and elegantly executed study. It's an in vitro study and therefore it's not clear how the observations relate to DNA repair in vivo. XPF-ERCC1, RPA and SNM1A are all involved in ICL repair and in cleavage of stalled forks, but the mechanisms behind how these factors cooperate to allow unhooking or fork cleavage in intact mammalian cells are unknown. Part of the reason for the lack of information in this area - and the reason why it's difficult to test if the findings in this paper are relevant to DNA repair in vivo - is that tools don't currently exist to allow a high-resolution view of DNA repair in intact mammalian cells. The McHugh study represents a valiant attempt to delineate a possible mechanism behind unhooking by XPF-ERCC1 by reconstituting

unhooking in vitro. The data provide a framework to think about fork cleavage and ICL repair that will hopefully be possible to test in more detail in the future.

Specific points:

1. This reviewer was confused at the effect of the nascent leading strand on XPF-ERCC1 activity. This is because many labs have reported that mammalian XPF-ERCC1 efficiently cleaves 3' flap substrates. There is strong evidence that yeast Rad1-Rad10 efficiently cleaves 3' flaps in vitro and also in vivo: during SSA repair of DSBs, the substrate of XPF-ERCC1 is clearly a 3' flap. So why would XPF-ERCC1 not cleave the 3' flap on the simple fork that has the nascent leading strand? This issue must be addressed thoroughly and comprehensively throughout the text, from the outset, to remove any potential confusion to readers.
2. The models presented from the current literature for how ICL repair works are very heavily biased towards the work from Johannes Walter's lab. Work from the Walter lab has allowed a high-resolution view of repair of an ICL on a small plasmid, where the ICL is encountered by two forks only because it's a small plasmid where that happens to be the case. Work from the Seidman lab indicates that two forks colliding with an ICL happens infrequently; this in no way takes from the work of the Walter lab, but it means that statements such as "ICL repair is triggered by the convergence of dual replication forks" are not appropriate because they imply that this is always the case. The available evidence argues against this idea in intact mammalian cells - other mechanisms are at play too. All of the data from McHugh and colleagues in this study seems to be interpreted in the context of the Walter model, and it seem sensible to modify the text so this is not the case.
3. It would be interesting to know if the physical interaction of XPF-ERCC1 with RPA is required for the effects of RPA seen in the experiments presented. If mutations are already known that abolish the interaction, then these should be tested. If not, then the lack of this information is not an obstacle to publication.

Referee #3:

ERCC1-XPF has been shown to play a key role in replication-coupled ICL repair. In the xenopus cell-free ICL repair system, ERCC1-XPF makes the first incision after a replication fork stalls at an ICL. An open question is how this incision is regulated and whether ERCC1-XPF makes incisions on both sides of the ICL or whether a second nuclease cuts on one side of the ICL.

In the present manuscript Adbullah and colleagues test the influence of RPA on the activity of ERCC1-XPF on model structures mimicking ICL repair intermediates. Strikingly, they show that RPA stimulates the activity of ERCC1-XPF specifically on substrates that mimic an approaching leading strand, but not on lagging strands. This stimulation was found to be dependent on a ssDNA binding surface on the lagging strand. In substrates containing site-specific ICLs, RPA directs the activity of ERCC1-XPF on the far side of the ICL and furthermore allows the loading of the exonuclease SNM1A, which can digest across the ICL, leading to complete unhooking of the ICL.

This study clearly advances our mechanistic understanding of how ERCC1-XPF, RPA and SNM1A function together in ICL repair. The ability of RPA to enable ERCC1-XPF to incise ICL repair intermediates in the exact situation predicted to arise in ICL repair is particularly striking. This biochemical study is innovative in its design and contains all the necessary controls to justify the conclusions drawn from the experiments.

After addressing the points listed below, I would consider it to be highly suitable for publication in EMBO J.

1. Wood and coworkers showed that ERCC1-XPF can make an incision on both sides of an ICL in a stem-loop substrate containing a psoralen ICL (Kuraoka, JBC, 2000, 275, 26632). In this paper, ERCC1-XPF is shown to first cut on the near side of the ICL (at the junction), followed by an incision on the other side of the ICL. This mechanism of unhooking differs from the one proposed in the present work, although that study did of course nor include RPA or SMN1A. Nonetheless, the authors should address this difference in the discussion. Is it perhaps due to the difference in the

structure of the ICL or the design of the substrate?

2. Another study, Sancar and coworkers showed that RPA can stimulate the activity of ERCC1-XPF to degrade DNA past an ICL, although this study employed duplex DNA (Mu et al, MCB, 2000, 20, 2446). As in point 1, this discrepancy should be addressed in the discussion. Both papers should also be cited in the introduction.

3. The structure of the ICL used here is quite different and likely less distorting than ICLs formed by the clinically more relevant cisplatin and nitrogen mustards, which have also been used to show the cellular involvement of ERCC1-XPF in ICL repair. A brief (speculative) discussion of how ICL structure might affect the activity of ERCC1-XPF would be warranted.

4. Figure 3. It would be instructive to compare the binding activity of RPA on a 3' Flap/lagging strand structure. This would reveal whether a difference of binding affinity or directionality of RPA binding is responsible for the selective stimulation of ERCC1-XPF by RPA. The later mechanism is suggested by DeLaat, 1998b, and it would be nice to make this connection here.

5. p11. Lane 7. Clarify the importance of why the use of a 5' OH group is important here. Is SNM1A only active on 5' phosphorylated substrates? If so, also clarify that ERCC1-XPF incision will leave a 5' phosphorylated product suitable for SNM1A to act on.

6. p15. Regarding FAN1: Although SNM1A and FAN1 share the ability to degrade a duplex past an ICL, they likely do so in different branches of ICL repair. The "redundancy" is therefore likely due to the usage of different pathways, rather than activity on the same substrates. This point could be clarified here.

7. Figures. It would be good to number to all lanes in the gels.

1st Revision - authors' response

13 April 2017

Many thanks for sending us the reviewers' comments for our manuscript '*RPA activates XPF-ERCC1 to initiate the processing of DNA interstrand crosslinks*'. We were delighted by the positive reviews. We have now addressed the vast majority of the points raised by the reviewers, and a detailed point-by-point response is attached to the resubmission. We hope the manuscript will now be acceptable for publication in EMBO J.

POINT-BY-POINT RESPONSE

Referee #1:

The enzymology of interstrand cross-link repair has not been well worked out. This biochemical study from an excellent team of scientists nicely demonstrates that XPF-ERCC1, one of the major nucleases known to be involved in interstrand cross-link repair is inhibited at a replication fork bearing a leading strand. Such a structure is expected to occur when the leading strand synthesis runs into an interstrand cross-link. This inhibition is shown to be overcome by the presence of the trimeric single-strand binding protein, RPA. Mechanistic studies suggest that this stimulation of XPF-ERCC1 incision is due to the binding of RPA70 subunit to the single-strand DNA end in the replication fork. They nicely show that the C-terminus of RPA70 is required. Using an elegant click chemistry approach to produce site-specific crosslinks at the site of the replication fork the authors then show that 1) XPF-ERCC1 cuts 6 nucleotides from the junction and this nick can serve as an entry point for the SNM1A exonuclease that can process through the cross-link. The authors provide a plausible model for interstrand cross-link repair at a single or a dual replication fork that is consistent with all known data. This is an important contribution to the field and makes direct predictions about the role of these proteins during repair in a living nucleus, which is clearly beyond the scope of this elegant work. The authors should consider the following points:

1. The small diagrams that are given with the incision experiments are extremely helpful. But in Figures 1, 2 and 3 showing the positions of the two nick sites at either 6 or 2 bp from the fork with

brackets is very confusing. Perhaps they can adopt the convention shown in Figure 5 and the length of the arrow can be proportional to the amount of cutting observed?

This has been done for all the relevant panels in Figs 1-3.

2. Figure 1 panel F, plotting the remaining product with the amount of incision product is helpful. But perhaps two Y axis could be shown, one indicating product remaining and the other incision products. The word "Formation" is very misleading as the uncut product is not forming.

This is a good suggestion, and has been done.

3. The authors show a direct interaction of XPF-ERCC1 with RPA in their model in Figure 6. Data supporting whether this direction interaction is necessary or sufficient versus the need for RPA binding to the DNA would provide impact. One experiment is to vary the length of this single strand DNA of the free end in the replication fork and determine the minimum length necessary for the stimulation. This type of experiment would suggest that binding of RPA to the single strand region allows interaction with XPF-ERCC1. I believe RPA requires 6 bases. Having the minimal flap size would then allow a fluorescence anisotropy experiment of XPF-ERCC1 (in the absence of Mg to inhibit incision) in the presence and absence of RPA. Binding of XPF-ERCC1 should show a large increase in anisotropy. A less elegant approach would be to do a pull down showing that XPF-ERCC1 can only load on a cross-link replication fork with a leading strand, if RPA is added. This could be done with biotin-labeled DNA and SA beads.

We agree that testing minimal flap size for XPF-ERCC1 stimulation by RPA is a good idea. This has been done, and the data presented in new FigEV3B and associated text. This revealed that RPA was not able to stimulate the activity of XPF-ERCC1 on a simple fork structure with 13 nt arms. However, a caveat here is that XPF-ERCC1 is substantially less active on the shorter arm substrates. The reviewer is correct that RPA can associate with ssDNA regions as short as 6 nt, but its high-affinity mode of binding requires 20-30 nt and is associated with a conformational change in the RPA trimer, and this appears to be required for the stimulatory effects observed here. Note that the helpful suggestion of reviewer 3 in regard of testing the affinity of RPA for the model lagging strand substrate (new FigEV4B) supports the data obtained by de Laat and colleagues almost 20 years ago that the orientation of RPA relative to XPF-ERCC1 is critical to its stimulatory role: it is not just efficient binding that is critical. In regards of the affinity of XPF-ERCC1 for the fork structures we have employed – this is many orders of magnitude less than that of RPA and anisotropy experiments have not proved very informative in our hands, although we continue to probe the mechanism of stimulation and whether it relates to increased binding efficiency or allosteric (or related) activation of XPF-ERCC1. Unravelling this is a major question that we think beyond the scope of the current paper.

4. Another nice experiment that would provide more support for their model, would be to mutate key residues of the XPF-ERRC1 or RPA that are suggested to be at the interface of their interaction site.

We have attempted to produce the version of XPF-ERCC1 that was previously proposed to contain a point mutation that disrupted interaction with RPA (P85S) (Fisher et al, JMB, 2011). This protein was insoluble, forming aggregates in expressing insect cells, possibly explaining why it was identified as a non-interactor in yeast two-hybrid studies. Notably, recent work from the Knipscheer lab (Klein-Douwel et al, EMBO J, 2017; now cited in our paper) indicates that the Xenopus XPF protein is also extremely sensitive to relatively small changes in the N-terminus. This raises the possibility that an interaction mediated by the N-terminus (possibly with RPA, which is highly conserved in the eukaryotic expression systems used) is important for maintaining the protein in an appropriate conformation in cells.

Referee #2:

In this paper, the McHugh and coworkers investigate the influence of the single-strand binding protein RPA on the ability of XPF-ERCC1 to unhook replication fork-blocking interstrand crosslinks. Taking a biochemical approach, using radiolabelled model replication fork structures,

they find that the addition of a nascent strand (either leading or lagging) to a "simple fork" (or "splayed Y") structure inhibits cleavage of the fork. They show that increasing nascent strand length increases the inhibitory effect on XPF-ERCC1. Intriguingly RPA, which is known to interact with XPF-ERCC1 and to facilitate the DNA repair processes that involve XPF-ERCC1, overcomes the inhibitory effect of nascent leading DNA strands on fork cleavage by XPF-ERCC1. RPA does not overcome the inhibitory effect of a nascent lagging strand, however. RPA appears to bias the position of incisions by XPF away from the fork junction. Evidence is presented that an intact RPA trimer is required to promote XPF-ERCC1 activity towards forks with nascent strands. The authors go on to show that placing an ICL at the fork junction biases the position of incisions by XPF away from the fork junction. This incision creates a site for loading of the SNM1A nuclease which can exonucleolytically degrade DNA past the ICL, thereby unhooking to ICL.

This an interesting and elegantly-executed study. It's an in vitro study and therefore it's not clear how the observations relate to DNA repair in vivo. XPF-ERCC1, RPA and SNM1A are all involved in ICL repair and in cleavage of stalled forks, but the mechanisms behind how these factors cooperate to allow unhooking or fork cleavage in intact mammalian cells are unknown. Part of the reason for the lack of information in this area - and the reason why it's difficult to test if the findings in this paper are relevant to DNA repair in vivo - is that tools don't currently exist to allow a high-resolution view of DNA repair in intact mammalian cells. The McHugh study represents a valiant attempt to delineate a possible mechanism behind unhooking by XPF-ERCC1 by reconstituting unhooking in vitro. The data provide a framework to think about fork cleavage and ICL repair that will hopefully be possible to test in more detail in the future.

Specific points:

1. This reviewer was confused at the effect of the nascent leading strand on XPF-ERCC1 activity. This is because many labs have reported that mammalian XPF-ERCC1 efficiently cleaves 3' flap substrates. There is strong evidence that yeast Rad1-Rad10 efficiently cleaves 3' flaps in vitro and also in vivo: during SSA repair of DSBs, the substrate of XPF-ERCC1 is clearly a 3' flap. So why would XPF-ERCC1 not cleave the 3' flap on the simple fork that has the nascent leading strand? This issue must be addressed thoroughly and comprehensively throughout the text, from the outset, to remove any potential confusion to readers.

The question of how XPF-ERCC1 cleaves 3'-flaps during SSA (a structure equivalent to the case where we have a nascent lagging strand in our model fork substrates) is indeed of interest. It is clear from some relatively old biochemical work we have cited (Rodriguez et al, JBC, 1996), which is consistent with the work we present here, that SSA-type flap intermediates are also much less efficiently cleaved than simple fork type structures by yeast Rad1-Rad10. However, there is a strong requirement for a further Rad1-Rad10 binding factor (Saw1, not conserved in higher eukaryotes & not involved in yeast ICL repair) for SSA in yeast (Li et al, EMBO J, 2013), and Saw1 is required to permit efficient Rad1-Rad10 cutting in the context of SSA intermediates. How XPF-ERCC1 in mammals catalyses incisions for SSA is not clear, since XPF-ERCC1 alone has very low activity on 3'-flaps, and is of interest, but beyond the scope of this study, but clearly will require additional factors.

2. The models presented from the current literature for how ICL repair works are very heavily biased towards the work from Johannes Walter's lab. Work from the Walter lab has allowed a high-resolution view of repair of an ICL on a small plasmid, where the ICL is encountered by two forks only because it's a small plasmid where that happens to be the case. Work from the Seidman lab indicates that two forks colliding with an ICL happens infrequently; this in no way takes from the work of the Walter lab, but it means that statements such as "ICL repair is triggered by the convergence of dual replication forks" are not appropriate because they imply that this is always the case. The available evidence argues against this idea in intact mammalian cells - other mechanisms are at play too. All of the data from McHugh and colleagues in this study seems to be interpreted in the context of the Walter model, and it seem sensible to modify the text so this is not the case.

This is completely fair, and an entirely new paragraph has been added into the Discussion to reflect the potential importance of the 'traverse' model.

3. It would be interesting to know if the physical interaction of XPF-ERCC1 with RPA is required for the effects of RPA seen in the experiments presented. If mutations are already known that

abolish the interaction, then these should be tested. If not, then the lack of this information is not an obstacle to publication.

Please see our comments to reviewer 1. We have tried to address this, but it remains elusive. Additional, perhaps structural, studies will be required.

Referee #3:

ERCC1-XPF has been shown to play a key role in replication-coupled ICL repair. In the xenopus cell-free ICL repair system, ERCC1-XPF makes the first incision after a replication fork stalls at an ICL. An open question is how this incision is regulated and whether ERCC1-XPF makes incisions on both sides of the ICL or whether a second nuclease cuts on one side of the ICL.

In the present manuscript Abdullah and colleagues test the influence of RPA on the activity of ERCC1-XPF on model structures mimicking ICL repair intermediates. Strikingly, they show that RPA stimulates the activity of ERCC1-XPF specifically on substrates that mimic an approaching leading strand, but not on lagging strands. This stimulation was found to be dependent on a ssDNA binding surface on the lagging strand. In substrates containing site-specific ICLs, RPA directs the activity of ERCC1-XPF on the far side of the ICL and furthermore allows the loading of the exonuclease SNM1A, which can digest across the ICL, leading to complete unhooking of the ICL.

This study clearly advances our mechanistic understanding of how ERCC1-XPF, RPA and SNM1A function together in ICL repair. The ability of RPA to enable ERCC1-XPF to incise ICL repair intermediates in the exact situation predicted to arise in ICL repair is particularly striking. This biochemical study is innovative in its design and contains all the necessary controls to justify the conclusions drawn from the experiments.

After addressing the points listed below, I would consider it to be highly suitable for publication in EMBO J.

1. Wood and coworkers showed that ERCC1-XPF can make an incision on both sides of an ICL in a stem-loop substrate containing a psoralen ICL (Kuraoka, JBC, 2000, 275, 26632). In this paper, ERCC1-XPF is shown to first cut on the near side of the ICL (at the junction), followed by an incision on the other side of the ICL. This mechanism of unhooking differs from the one proposed in the present work, although that study did of course not include RPA or SMN1A. Nonetheless, the authors should address this difference in the discussion. Is it perhaps due to the difference in the structure of the ICL or the design of the substrate?

This is a good point and is addressed in a new section of the Discussion, starting on page 15

2. Another study, Sancar and coworkers showed that RPA can stimulate the activity of ERCC1-XPF to degrade DNA past an ICL, although this study employed duplex DNA (Mu et al, MCB, 2000, 20, 2446). As in point 1, this discrepancy should be addressed in the discussion. Both papers should also be cited in the introduction.

Fair point - these papers are now cited and discussed in the Introduction.

3. The structure of the ICL used here is quite different and likely less distorting than ICLs formed by the clinically more relevant cisplatin and nitrogen mustards, which have also been used to show the cellular involvement of ERCC1-XPF in ICL repair. A brief (speculative) discussion of how ICL structure might affect the activity of ERCC1-XPF would be warranted.

Our ICLs are indeed relatively non-distorting. We have combined discussion of this point (page 15) with that relating to point 1 by this reviewer as they are linked issues.

4. Figure 3. It would be instructive to compare the binding activity of RPA on a 3' Flap/lagging strand structure. This would reveal whether a difference of binding affinity or directionality of RPA binding is responsible for the selective stimulation of ERCC1-XPF by RPA. The later mechanism is suggested by DeLaat, 1998b, and it would be nice to make this connection here.

This is an excellent experimental suggestion, which has been performed, and presented at new FigEV4B and associated text. This reveals that RPA has a similar affinity for the ssDNA regions of model forks bearing nascent leading or lagging strands, implying that we are observing an orientation/polarity for the RPA stimulation of XPF-ERCC1, as first identified by de Laat et al in 1998 using substrates relevant to NER, and agree that this connection and corroboration of data is very satisfying.

5. p11. Lane 7. Clarify the importance of why the use of a 5' OH group is important here. Is SNM1A only active on 5' phosphorylated substrates? If so, also clarify that ERCC1-XPF incision will leave a 5' phosphorylated product suitable for SNM1A to act on.

We have clarified this point in the text – yes, SNM1A requires a 5'-phosphate to act.

6. p15. Regarding FAN1: Although SNM1A and FAN1 share the ability to degrade a duplex past an ICL, they likely do so in different branches of ICL repair. The "redundancy" is therefore likely due to the usage of different pathways, rather than activity on the same substrates. This point could be clarified here.

Fair point, we have clarified this in the relevant section of the Discussion.

7. Figures. It would be good to number to all lanes in the gels.

This has been done.

2nd Editorial Decision

15 May 2017

Thank you for submitting your revised manuscript for our editorial consideration. It has now been once more assessed by two of the original referees, who are both satisfied with the revisions and fully supportive of publication now (see comments below). We shall therefore be happy to publish your study in The EMBO Journal at this point!

Referee #1 (Report for Author)

I downloaded the revised manuscript and the rebuttal letter and have gone through them both carefully.

The study has been greatly strengthened by the review process. The authors have nicely dealt with all the concerns raised by each reviewer, in several instances performing new experiments to better discern the mechanism they are observing. These additional experiments and stronger citation of the pertinent literature strengthen the already excellent study. I believe this study will have a significant impact on the field and is worthy of a highlight in your journal.

Referee #3 (Report for Author)

The authors have satisfactorily addressed the concerns raised by the reviewers. Figures EV3B and EV4B add to the quality of an already excellent manuscript and I am in agreement with the argument of the authors in response to reviewer 2 that the question of why ERCC1-XPF has low activity on 3' flaps is a complex issue better addressed in a different manuscript. All points regarding data presentation, the discussion and citation of references has also been adequately addressed.

This manuscript is now suitable for publication in EMBO J.

YOU MUST COMPLETE ALL CELLS WITH A PINK BACKGROUND

Corresponding Author Name: Peter J. McHugh
Journal Submitted to: EMBO J
Manuscript Number: EMBOJ-2017-96664